# Systems Biology of Recombinant 2G12 and 353/11 mAb Production in CHO-K1 Cell Lines at Phosphoproteome Level

**DOI:** 10.3390/proteomes13010009

**Published:** 2025-02-10

**Authors:** Eldi Sulaj, Felix L. Sandell, Linda Schwaigerlehner, Gorji Marzban, Juliane C. Dohm, Renate Kunert

**Affiliations:** 1Department of Biotechnology and Food Science, Institute of Animal Cell Technology and Systems Biology (IACTSB), BOKU University, Muthgasse 18, 1190 Vienna, Austria; eldi.sulaj@boku.ac.at (E.S.); linda.schwaigerlehner@gmail.com (L.S.); renate.kunert@boku.ac.at (R.K.); 2Department of Biotechnology and Food Science, Institute of Computational Biology (ICB), BOKU University, Muthgasse 18, 1190 Vienna, Austria; felix.sandell@boku.ac.at (F.L.S.);; 3Department of Biotechnology and Food Science, Institute of Bioprocess Science and Engineering (IBSE), BOKU University, Muthgasse 18, 1190 Vienna, Austria

**Keywords:** Chinese hamster ovary cell, phosphoproteome, tunicamycin, LFQ, mass spectrometry

## Abstract

**Background**: Chinese hamster ovary (CHO) cells are extensively used in the pharmaceutical industry for producing complex proteins, primarily because of their ability to perform human-like post-translational modifications. However, the efficiency of high-quality protein production can vary significantly for monoclonal antibody-producing cell lines, within the CHO host cell lines or by extrinsic factors. **Methods**: To investigate the complex cellular mechanisms underlying this variability, a phosphoproteomics analysis was performed using label-free quantitative liquid chromatography after a phosphopeptide enrichment of recombinant CHO cells producing two different antibodies and a tunicamycin treatment experiment. Using MaxQuant and Perseus for data analysis, we identified 2109 proteins and quantified 4059 phosphosites. **Results**: Significant phosphorylation dynamics were observed in nuclear proteins of cells producing the difficult-to-produce 2G12 mAb. It suggests that the expression of 2G12 regulates nuclear pathways based on increases and decreases in phosphorylation abundance. Furthermore, a substantial number of changes in the phosphorylation pattern related to tunicamycin treatment have been detected. TM treatment affects, among other phosphoproteins, the eukaryotic elongation factor 2 kinase (Eef2k). **Conclusions**: The alterations in the phosphorylation landscape of key proteins involved in cellular processes highlight the mechanisms behind stress-induced cellular responses.

## 1. Introduction

Chinese hamster ovary (CHO) cells are the leading mammalian cell culture system in biomanufacturing, used in 95 of 107 distinct biopharmaceutical products, making up about 89% of all mammalian cell-derived products [1]. The analysis of the cellular proteome has enabled researchers to precisely identify, study, and quantify thousands of proteins from complex samples [2]. This approach bridges the gap to genomic data, offering insights into how genes translate into protein expression and elucidating the activity within biological systems [2]. The first proteome study in CHO-K1 cells identified over 6000 expressed proteins [3] and since then we have profoundly enhanced our understanding of protein expression dynamics in CHO cells [4,5,6,7,8,9,10,11,12,13,14,15,16,17]. Furthermore, the analysis of the cellular phosphoproteome will provide essential information on protein regulation, signaling pathways, and cellular responses to various stimuli through the analysis of phosphorylation sites [18].

Given the importance of CHO cells, a deeper understanding of their molecular mechanisms is essential, particularly in analyzing phosphoproteome alterations caused by the production of complex biomolecules like monoclonal human antibodies, which remain inadequately explored. To our knowledge, there are a limited number of published studies on phosphoproteomics approaches in CHO cells [10,13,19,20,21,22].

Protein phosphorylation is a key post-translational modification (PTM) and plays a cardinal role in regulating, switching on/off, and modulating most biological processes [23,24]. It is projected that approximately 30% of all proteins expressed in eukaryotic cells are phosphorylated at some point during their existence [25,26]. Nevertheless, only about 2–3% of all eukaryotic genes are believed to encode protein kinases [27,28].

The identification of the phosphoprotein vitellin in 1906 by [29], its first isolation and purification by [30], and first demonstration of protein kinase activity [31], marked a pivotal advancement in the field. Kinases catalyze the transfer of the gamma-phosphate from ATP or GTP to the acceptor hydroxyl residue, typically serine, threonine, or tyrosine, of a protein substrate [32,33]. This process, known as phosphorylation, is crucial for regulating various cellular activities, including metabolism, transcription, cell cycle progression, cytoskeletal rearrangement and cell movement, apoptosis, and differentiation [34]. Phosphorylation and dephosphorylation of serine, threonine, and tyrosine residues are ubiquitous and fundamental in eukaryotic signaling [35,36]. The attachment of a phosphate group leads to an increase in protein mass of 80 daltons (the added mass of the phosphate group) [37,38,39,40] or multiples of 80 Da for multiple sites [41,42], but the low stoichiometric [43] or substoichiometric abundance [44] of many phosphopeptides demands phosphopeptide enrichment to distinguish them from the high concentrations of non-phosphorylated peptides.

The main aim of our study was to explore the phosphorylation status among CHO cells producing recombinant difficult-to-produce 2G12 vs. easy-to-produce 353/11, as well as to examine the influence of the extrinsic stressor tunicamycin (TM) on the protein expression. Therefore, we performed a comparative analysis of the phosphoproteomic profiles utilizing label-free quantification by mass spectrometry (LFQ-MS) of two CHO cell lines that express two different monoclonal antibodies (difficult-to-produce 2G12 and easy-to produce 353/11), as well as tunicamycin-treated 353/11 CHO cell line and untreated 353/11 CHO cell line (Figure 1). The effects of tunicamycin (TM), a compound that is known to inhibit *N*-linked glycosylation, [45,46] were additionally analyzed as a research tool to induce endoplasmic reticulum (ER) stress and to activate the unfolded protein response (UPR) [47,48,49]. It functions as an inhibitor of N-acetylglucosamine-1-phosphate transferase (GPT), thereby obstructing the initial step of glycoprotein biosynthesis [50].

In a prior study, we performed an extensive proteomic analysis to establish a baseline understanding of the cellular proteome [12], recognizing beforehand that the interpretation of quantitative phosphoproteomics is a composite of both protein expression and phosphorylation dynamics [51]. The results showed an increase in the abundance of proteins associated with protein folding mechanisms in the low producer compared to the high producer cell line. Further, Hspa9 and Dnaja3 were observed as candidates activated by the mitochondria UPR and play important roles in protein folding processes in mitochondria. We identified a significant increase in the abundance of Nedd8 and Lgmn proteins in similar levels which may contribute to UPR stress. Our findings elucidated that in the first comparison between 2G12 (low producer) and 353/11 (high producer), monoclonal antibody production was delineated as an intrinsic factor, whereas in the second comparison, 353/11_TM (high producer treated with TM) and 353/11 (high producer), tunicamycin functioned as an extrinsic factor influencing the modulation of protein production. Building upon this foundation, our current research aims to understand the differences in phosphoproteomics profiles between high and low producer cell lines. Specifically, we aim to identify and compare phosphorylation patterns that contribute to variations in cellular productivity. Furthermore, we investigate how tunicamycin-induced endoplasmic reticulum (ER) stress influences cellular responses, with a specific focus on the unfolded protein response (UPR). This analysis aims to reveal the molecular mechanisms governing cellular responses to ER stress and their implications for protein production and overall cellular function at the phosphorylation level.

## 2. Materials and Methods

### 2.1. Cell Lines and Sample Preparation

The creation of isogenic CHO cell lines for producing human IgG_1_ antibodies was previously described by Schwaigerlehner et al. [52]. Additionally, a comprehensive experimental setup for analyzing the cellular proteome was provided in a subsequent study by Sulaj et al. [12]. The cells were maintained in chemically defined CD-CHO medium (Gibco, no. 10743-029, Grand Island, NY, USA), supplemented with 4 mM L-glutamine (Roth, no. 9183.1, Karlsruhe, Germany), 15 mg/L phenol red (Sigma-Aldrich, no. P0290, Schnelldorf, Germany), and 2 μM ganciclovir (GCV) (Sigma-Aldrich, no. G2536-100MG, Schnelldorf, Germany). As described in [12,52], cells were seeded at an initial density of 5 × 10^6^ cells/mL and grown in suspension culture using a semi-perfusion method with daily medium refreshment by pelleting of cells and adding new medium [53,54]. Cells were cultured in 50 mL vent cap spin tubes (Corning, No. 431720, Corning, NY, USA) in an ISF-X shaker (Kühner, Basel, Switzerland) at 37 °C, 80% humidity, 5% CO_2_, and 220 rpm. We used this downscaling procedure to simulate a continuous biological process, the “healthiest” and most natural cultivation conditions for biological systems.

In general, the perfusion bioprocess is a cultivation method that promotes physiological conditions close to those of the animal organism by supplying nutrients and removing negative waste products, such as lactate and ammonium. Although many studies have attempted to investigate the difference between exponential and steady-state growth phases in batch cultures, the influence of extrinsic factors resulting from the non-physiological conditions remains evident. We have shown in two papers that extrinsic conditions have a massive effect on phenotype [55] and gene transcription [56]. The fact that animal cells are able to change their transcriptional behavior in a moderate time frame led us to an experimental approach in which cells were harvested four hours after the daily media change to ensure physiological homeostasis and to guarantee the same extrinsic conditions for tunicamycin-treated and untreated cells. The conditions of tunicamycin treatment (only 4 h at a low concentration) were chosen to identify the differences that play a role in the initiation-phase of the stress response, while avoiding the induction of a stronger ER stress response which is found to be characterized by the pronounced upregulation of well-known proteins, including Hsp90b1, Pdia4, and Grp78 (Hspa5) [57]. This approach is expected to facilitate the identification of early regulatory mechanisms, in contrast to the strong ER stress responses induced by higher tunicamycin concentrations. Both cell lines achieved maximum cell density and sustained viability by day six. Samples were collected four hours after the medium exchange on that day, when cells had reached maximum density.

Among the cell lines, 353/11 exhibited high production levels, contrasting with lower production levels observed in 2G12. The data highlighting these differences are described in a prior publication from our research group [52]. Cell line 353/11 was treated separately with tunicamycin (TM) (Sigma-Aldrich, no. T7765, Schnelldorf, Germany) for the induction of endoplasmic reticulum (ER) stress. To prepare phosphoproteomics samples, cell cultures were harvested on day six. At the time of sampling, the average viable cell density reached 4 × 10^7^ cells/mL, with viabilities consistently maintained at approximately 99%. Samples were harvested 4 h after medium exchange to synchronize sampling across all cultivars and for inducing mild stress, one set of samples (353/11) was treated with 1 µg/mL TM in fresh medium also for four hours. Cell suspensions were centrifuged at 1300 rpm for 7 min at room temperature, washed twice in PBS buffer, and stored at −80 °C.

### 2.2. Protein In-Solution Digestion and Peptide Desalting

To prepare for in-solution protein digestion, samples were processed as described previously [12]. Each cell pellet, consisting of 2 × 10^6^ cells, was resuspended in a 200 µL solution composed of 50 mM tetraethylammonium bromide (TEAB) (Sigma-Aldrich, no. T7408, Schnelldorf, Germany) and 8 M urea buffer (pH 8.0) (Sigma-Aldrich, no. 51456, Schnelldorf, Germany). The mixture was transferred to a new low protein binding microcentrifuge tube (Thermo Fisher Scientific, no. 88379, Rockford, IL, USA), followed by the addition of 2 µL protease and phosphatase inhibitor cocktail (EDTA-free) (Thermo Fisher Scientific, no. 78443, Rockford, IL, USA).

The samples underwent lysis in an ice-cold Branson 2510 ultrasonic bath (Emerson Electric, Saint Louis, MO, USA) for 30 min at a frequency of 40 kHz. Afterwards, the samples were diluted with an additional 200 µL of 50 mM TEAB and centrifuged at 2500× *g* for 30 min at 4 °C. The protein concentration of the supernatant was determined using the Micro BCA^TM^ Protein Assay Kit (Thermo Fisher Scientific, no. 23235, Rockford, IL, USA). Reduction was achieved by adding 2 µL of 1 M dithiothreitol (DTT) (Thermo Fisher Scientific, no. 20291, Rockford, IL, USA) and incubating at 37 °C for 60 min on a rotating shaker. The samples were then alkylated with 500 mM iodoacetamide (IAA) (Thermo Fisher Scientific, no. 90034, Rockford, IL, USA) for 30 min at 27 °C in the dark, followed by quenching with 414 µL of 50 mM TEAB.

For in-solution digestion, 15 µL (at a 30:1 protein/protease *w*/*w* ratio) of Trypsin/Lys-C Protease Mix, MS Grade (Thermo Fisher Scientific, no. A40009, Rockford, IL, USA) was added and the samples were incubated overnight on a shaker at 37 °C. Finally, the digestion process was terminated with the addition of 25% trifluoroacetic acid (TFA) (Thermo Fisher Scientific, no. 28904, Rockford, IL, USA) to achieve a final concentration of 0.2% TFA in the mixture. The resulting peptides were rapidly dried using a Concentrator plus speed vacuum system (Eppendorf, Hamburg, Germany).

Peptide desalting was performed using Pierce^TM^ Peptide Desalting Spin Columns (Thermo Fisher Scientific, no. 89852, Rockford, IL, USA) per the manufacturer’s instructions. The columns were equilibrated by centrifugation at 5000× *g* for 1 min to remove the storage buffer, then equilibrated with two washes of 0.1% trifluoroacetic acid (TFA) in 80% acetonitrile (ACN), followed by two washes with 0.1% TFA. Peptides were reconstituted in 300 µL of 0.1% TFA, loaded onto the columns, and centrifuged at 3000× *g* for 1 min. After two washes with 0.1% TFA, phosphopeptides were eluted with 0.1% TFA in 80% ACN and vacuum dried.

### 2.3. Phosphopeptide Enrichment

Phosphopeptides were selectively enriched using the High-Select^TM^ TiO_2_ Phosphopeptide Enrichment Kit (Thermo Fisher Scientific, no. A32993, Rockford, IL, USA). To prepare the column, a centrifuge column adaptor was placed into a new 2 mL low protein binding microcentrifuge tube and a TiO_2_ spin tip was inserted into the adaptor. The column was conditioned with 20 μL of wash buffer and centrifuged at 3000× *g* for 2 min, followed by 20 μL of binding/equilibration buffer and another centrifugation at 3000× *g* for 2 min. To bind phosphopeptides, the equilibrated TiO_2_ spin tip and adaptor were transferred into a new 2 mL low protein binding microcentrifuge tube. Desalted, dried peptides were dissolved in 150 µL of binding/equilibration buffer and applied to the spin tip, then centrifuged at 1000× *g* for 5 min. The sample from the microcentrifuge tube was reapplied twice to the spin tip to increase phosphopeptide yield. Subsequently, the column was washed with 20 μL of LC-MS grade water (Thermo Fisher Scientific, no. 51140, Rockford, IL, USA) (autoclaved reverse osmosis (RO) water). The bound phosphopeptides were eluted using 50 µL of phosphopeptide elution buffer, with two rounds of centrifugation at 1000× *g* for 5 min each. Subsequently, the eluate was promptly vacuum dried.

### 2.4. LC-MS/MS Analysis

The dried phosphopeptide samples were resuspended in 25 μL of 0.1% formic acid (FA), LC-MS grade (Thermo Fisher Scientific, Cat. No. 85170, Rockford, IL, USA). For the nano LC, an UltiMate^TM^ 3000 RSLCnano System (Dionex, Thermo Fisher Scientific, Sunnyvale, CA, USA) was employed. The mass spectrometry analysis was conducted using an Orbitrap Q Exactive Plus^TM^ instrument (Thermo Fisher Scientific, IL, USA). Peptide samples were pre-concentrated using a µ-pre-column with dimensions of 300 µm inner diameter (I.D.), 5 µm particle size, and 100 Å pore size (Thermo Fisher Scientific, IL, USA). Separation was achieved using a 50 cm Acclaim PepMap^TM^ 100 C18 column (50 cm × 75 µm, 2 µm) (Thermo Fisher Scientific, IL, USA). The flow rate was maintained at 300 nL/min. Mobile Phase A consisted of ultrapure water with 0.1% (*v*/*v*) formic acid, and Mobile Phase B consisted of 80% acetonitrile with 0.08% (*v*/*v*) formic acid. The gradient was programmed to transition from 3% to 40% Mobile Phase B over 77 min, followed by an increase to 95% in 2 min, and maintained at 95% B for 17 min as a washing step.

The mass spectrometer (MS) was operated in positive ion mode utilizing data-dependent acquisition (DDA). The settings were as follows: spray voltage at 2 kV, capillary temperature at 275 °C, and a mass range of 375–1500 *m*/*z* with a resolution of 70,000 for MS spectra. The automatic gain control (AGC) target was set to 1e6 with a maximum ion accumulation time of 50 ms. For MS/MS spectra, up to the top 15 precursor ions (charge states 2–6) were selected with a resolution of 17,500. The dynamic exclusion duration was 50 s, and the isolation window was set to 2.0 *m*/*z*. Lock masses at 445.12003 *m*/*z* and 391.28429 *m*/*z* were used to ensure mass accuracy.

### 2.5. Data Processing LC-MS/MS

Label-free quantification (LFQ) of phosphoproteomics was analyzed using data files (RAW format) in MaxQuant software version 2.0.3.1. The workflow was based on a previously published paper [12] and adjusted according to guidelines provided by the software’s developers [58]. Samples were clustered into three sample groups: 2G12, 353/11, and 353/11_TM, with four biological replicates for each sample group. In MaxQuant, the following parameters were applied: the type was set to “standard” for LFQ; and multiplicity was set to “1” (indicating no isotopic labeling was used). Variable modifications included: oxidation of methionine (Oxidation (M)), N-terminal acetylation (Acetyl (Protein N-term)), and phosphorylation of serine, threonine, and tyrosine (Phospho (STY)). Fixed modification included: carbamidomethylation of cysteine (Carbamidomethyl (C)). The digestion mode was set to “Trypsin/P” with a maximum of two missed cleavages. The minimum number of peptides required for a protein to be included in pairwise comparisons between samples was set to two (min. ratio count: 2) and the normalization type was set to “classic”. To ensure stringent identification confidence, a false discovery rate (FDR) of 1% was applied to peptide spectral matches (PSM), site decoy matches, and protein and peptide identifications. MS/MS spectra were searched using MaxQuant with its proprietary peptide database search engine Andromeda [59], against databases comprising 58,083 entries for *C. griseus* (UniProtKB id: 10029, downloaded on 12 April 2023) and 88534 entries for *M. musculus* (UniProtKB id: 10090, downloaded on 12 April 2023). The MS phosphoproteomics data have been deposited to the ProteomeXchange Consortium via the PRIDE [60] partner repository with the dataset identifier PXD055200.

### 2.6. Bioinformatics Analysis

The statistical analysis was performed using Perseus platform version 2.0.6.0 following the detailed guidelines outlined in the original publications by its developers [61,62]. We filtered the “Phospho (STY)Sites.txt” file to remove phosphosite entries labeled as “reverse” or “potential contaminants”, and we retained only Class I phosphosites with a probability score greater than 0.75 (localization probability filter > 0.75), indicating a high confidence score measuring the accuracy of identifying and localizing a specific phosphorylation site on a peptide. With the “expand site” function, data were log2-transformed (Appendix A). A categorical annotation was implemented to classify the data columns into their respective groups: 2G12, 353/11, and 353/11_TM. We required a minimum of three non-missing values per site in at least one sample group. Sites that did not meet this criterion were excluded. Missing values were imputed using the “replace missing values from normal distribution” function, in total matrix mode, applying a downshift of 1.8 times the standard deviation and a width of 0.3 times the standard deviation of the dataset.

Following the successful identification of phosphosites, a two-sample *t*-test was performed between the columns corresponding to the groups 2G12 vs. 353/11 and 353/11_TM vs. 353/11. The statistical significance of differentially regulated phosphosites was determined by applying a Benjamini–Hochberg False Discovery Rate (BH–FDR) correction, with a significance threshold set at *q* < 0.05 (Appendix A). Appendix A is a refined subset of Appendix A, derived from the “Phospho (STY) Sites.txt” output file from MaxQuant and includes only the phosphosites identified as significant in our comparative analysis of sample groups representing different cell lines and TM treatments. Each entry ID (phosphosite) in the dataset was further characterized using the Perseus platform, incorporating gene annotations from multiple sources, including Gene Ontology (GO), Kyoto Encyclopedia of Genes and Genomes (KEGG), Pfam, Gene Set Enrichment Analysis (GSEA), Keywords, CORUM and other UniProt databases (Appendix A). This annotation process leveraged databases for *C. griseus* and *M. musculus* to maximize the comprehensiveness of annotations across the dataset.

We analyzed the gene ontologies of the 74 differentially expressed phosphosites using ShinyGO 0.80 [63] (Appendix A), with the ten most statistically significant terms being afterward visualized. The x-axis of the resulting plots represents the ratio of enriched genes associated with a specific GO term to the total number of genes in the organism annotated with that term. The results were cross-validated through a literature check with the help of other enrichment analysis tools, such as the Database for Annotation, Visualization, and Integrated Discovery as described before [64,65], (DAVID version 2021) and Metascape [66]. We performed term annotations using ShinyGO with STRING v11.5, applying the *M. musculus* database for the 353/11_TM vs. 353/11 comparison and the *C. griseus* database for the 2G12 vs. 353/11 comparison. Due to fewer GO annotations for *C. griseus* compared to *M. musculus*, this selection of databases was designed to increase the rigor of our analyses, particularly for tunicamycin, to provide more detailed insights. Furthermore, network analysis of the most significantly enriched terms associated with phosphoproteins across the two comparisons was performed (Appendix A). This analysis involved constructing and visualizing networks to illustrate the relationships and functional associations among the top enriched terms using a false discovery rate (FDR) cut-off of 0.05 and an edge cut-off of 0.3 via the ShinyGO tool. The top 15 terms were selected based on their FDR values, with the “remove redundancy” option applied to remove similar terms. For both tables (Appendix A), the analysis was performed using the *C. griseus* database. By integrating and mapping these terms, we aimed to uncover underlying biological themes and interactions within the phosphoproteomic data, providing a comprehensive understanding of the functional implications of the identified phosphoproteins.

### 2.7. Computing Resources

Data processing was conducted using a high-performance Linux computing cluster comprising seven nodes running the CentOS 6.7 and CentOS 7 operating systems (Copyright 2024 The CentOS Project; https://www.centos.org/). Each node was equipped with either 24 cores at 2.6 GHz or 32 cores at 3.3 GHz and supported a maximum RAM capacity of 1 TB.

## 3. Results and Discussion

### 3.1. Quantative Phosphoprotemics Profiling Reveals Diferences

The phosphoproteomics workflow applied in this study is illustrated in Figure 1. The dynamic changes in the phosphoproteome and proteome of CHO cells remain largely unexplored and insufficiently characterized, with only a handful of studies performed each year over the past decade. To address this gap, we conducted a phosphoproteome analysis of mAb producing CHO-K1 cell lines, aiming to deepen our understanding of this regulatory network and contribute to the growing body of comparable data in this field. We identified 2109 proteins linked to 2504 distinct phosphorylated peptides and quantified 4059 phosphosites in our CHO samples (Appendix A). Our analysis highlights molecular differences between the 2G12 (low producer) and 353/11 (high producer) groups, recognizing monoclonal antibody production as an intrinsic factor for phosphoproteomic differences. Furthermore, we investigated the differences between the 353/11_TM (high producer treated with TM) and 353/11 (high producer) groups, where tunicamycin acts as an extrinsic factor influencing cellular homeostasis. The number of identified phosphosites aligns with results from other studies on phosphoproteins in CHO cells [19,22]. Among the quantified phosphosites, 1938 were single phosphorylation sites (multiplicity of 1), 1647 were double phosphorylation sites (multiplicity of 2), and 474 had three or more phosphorylation sites (multiplicity of 3). The varying degrees of phosphorylation suggest that different levels of phosphorylation can lead to distinct regulatory effects and biological outcomes [26,67].

The analysis of phosphorylated residues revealed that serine (S) is the most frequently phosphorylated amino acid, with 3744 phosphosites, followed by threonine (T) with 305 phosphosites, and tyrosine (Y) with 10 phosphosites. This distribution is attributable to the high prevalence of protein kinases specific to serine and threonine residues (Ser/Thr kinases) [68], which constitute 80% of known kinases [69]. Historical autoradiography measurements estimated the ratios of phosphorylated serine (S), threonine (T), and tyrosine (Y) residues as 90:10:0.05 [68,70]. In our analysis, we observed proportions of 92.20%, 7.51%, and 0.29% for S, T, and Y phosphosites, respectively, highlighting the predominant occurrence of serine and threonine phosphorylation compared to tyrosine phosphorylation within the dataset.

To examine the differences in phosphosites in the 2G12 vs. 353/11 and 353/11_TM vs. 353/11 comparison groups, a two-sample *t*-test was conducted, correcting for multiple comparisons using the Benjamini–Hochberg False Discovery Rate (BH–FDR) method, with a significance threshold of *q*-value < 0.05. Detailed statistical data for each phosphosite are provided in Appendix A. Notably, out of 4059 phosphosites across all three groups, 2G12, 353/11, and 353/11_TM, only 74 were statistically significant. Specifically, 53 significant phosphosites (associated with 41 phosphopeptides) were identified between the 2G12 (low producer) and 353/11 (high producer) groups, while 26 significant phosphosites (associated with 19 phosphopeptides) were found between the 353/11_TM (high producer treated with TM) and 353/11 (high producer) groups (Figure 2) (Appendix A).

To better depict the variations among the 74 significantly identified phosphosites, we performed Z-score normalization and then conducted hierarchical clustering for visualization in a heatmap (Figure 3). Rows within the same color cluster show similar phosphorylation levels, with colors showing either increases or decreases in z-scores, reflecting coordinated regulatory responses. Phosphosites represented in orange (increased Z-scores) indicate enhanced phosphorylation, whereas phosphosites shown in light blue (decreased Z-scores) signify reduced phosphorylation. The interplay of these different levels of phosphorylation not only serves as a critical mechanism for switching protein function but also introduces a layer of complexity through these phosphorylation sites, allowing for sophisticated combinatorial regulation and fine-tuning of protein activities and their associated modifications [71]. Particularly, the 2G12 group forms a distinct cluster in the upper left quadrant, characterized predominantly by lower occupancy levels for many phosphosites. This decrease in phosphorylation abundance distinguishes 2G12 from the other two groups, indicating a less active cellular environment. In contrast, the 353/11_TM group displays a pronounced increase in phosphorylation abundance, with many sites showing significantly higher abundance levels compared to 2G12. Phosphosites such as A0A8C2M783 (Srrm1) (serine/arginine repetitive matrix protein 1) and Q9QXP3 (high mobility group protein HMG-I/HMG-Y) show an increase in phosphorylation. The 353/11 group, while more similar to 353/11_TM, shows an intermediate expression profile, with a mix of high and low phosphorylation across different proteins. This suggests that 353_11 expression may represent a transitional state, in which cellular processes are moderately active or even more balanced than in the other groups but not as elevated as in 353/11_TM. Phosphosites shown in the heatmap provide valuable insights into the underlying molecular mechanisms specific to each group, defined by their correlation with protein synthesis and cellular regulatory changes induced by TM treatment.

### 3.2. Gene Ontology Enrichment Analyses

Enrichment analysis of phosphoproteins with altered abundance was performed using Gene Ontology (GO) analysis via ShinyGO (version 0.80) [63]. To ensure a focused analysis of the most relevant findings, we selected the top 10 most statistically significant terms from each comparison. These terms are presented in Figure 4 as a bar graph, describing the Gene Ontology (GO) terms. Data curation was facilitated by the use of several other tools, including the Database for Annotation, Visualization, and Integrated Discovery, as described before [64,65], (DAVID version 2021), and Metascape [66] to improve cross-referencing with existing literature and increase the robustness of the analysis. The respective tools are accessible online at (http://bioinformatics.sdstate.edu/go/), (https://david.ncifcrf.gov/tools.jsp) and (https://metascape.org). ShinyGO (version 0.80) was utilized according to the developers’ recommendations, with a false discovery rate (FDR) cutoff of 0.05, a minimum pathway size of 2, and annotation sources based on STRING [72] version 11.5 or Ensembl [73] release 104 databases. The data in Figure 4 are detailed in Appendix A.

To gain deeper insights into phosphoproteins and their biological effects, we performed the Gene Ontology (GO) analysis encompassing biological processes (BP), molecular functions (MF), and cellular components (CC). Figure 4 illustrates the GO analyses of both comparisons: the mAb2G12 expression (Figure 4a,c,e) as an intrinsic factor and the TM treatment as an extrinsic factor (Figure 4b,d,f). The length of each bar represents the gene ratio (nGenes/PathwayGenes) associated with the corresponding enriched term. The color coding reflects the varying degrees of statistical significance among the enriched terms, with blue indicating the highest significance (low FDR values) and orange representing relatively lower significance (higher, but statistically significant, FDR values).

Figure 4a,b and Appendix A show analyses focusing on “biological processes”. In the case of 2G12 as the factor of our comparison (2G12 vs. 353/11), 14 terms were identified as being regulated. Most of them are related to gene transcription, RNA maturation, and biochemical reactions and pathways involving long unbranched macromolecules formed from ribonucleotides. Studies suggest that phosphorylation may play a role and modulate protein–protein interactions within the spliceosome [74]. The main players identified in these terms and regulated by factor 2G12 expression are Hnrnpk, Taf2, Utp18, Rbm39, Srrm1, Farsa, Usp39, Nol11, Smarca4, Wdr55, Iws1, and Nolc1.

In the case of TM treatment as a factor in the comparison between 353/11_TM and 353/11, we found BP terms characterizing the TOR signaling pathway, the cellular development pathway, spine morphogenesis or cellular size, or growth of axons, as well as modification of phosphorylation by a protein as a modulator. These terms indicate a broad spectrum of cell biological changes induced by TM treatment with regulated phosphoproteins, such as Rps6, Larp1, Ehmt2, Dbnl, Hsp90ab1, Smarca4, Map1b, Nolc1, Eef2k, and Tnks1bp1. The most important terms related to TM treatment are “DNA methylation on cytosine within a CG sequence”, “DNA methylation on cytosine”, and “response to organonitrogen compound”.

It is worth noting that GO terms are heavily curated to make them better/clearer in the long run. Some of these terms have recently been moved to the “MF” category by the GO consortium, but still have statistical significance.

Further Gene Ontology (GO) analyses for molecular function (MF) related to each comparison are provided in the Appendix A and visualized in Figure 4c,d. The GO (MF) analysis of phosphoproteins in the comparison between the 2G12 and 353/11 samples revealed significant enrichment in several key molecular functions, including “basal transcription machinery binding”, “RNA binding”, “RNA polymerase core enzyme binding”, “nucleic acid binding”, and “transcription factor binding”. The GO analysis comparing the 353/11_TM and 353/11 samples demonstrated significant enrichment in “RNA binding”, “DNA polymerase binding”, “protein domain-specific binding”, “protein-containing complex binding”, and “nucleic acid binding”. The effect of TM treatment was found in a variety of contexts, including the binding of DNA polymerase and p53 family proteins, as well as phospholipids and altered cytoskeletal reorganization. This again suggests a more complex effect of TM compared to 2G12 as the determining factor.

Analyses of the cellular component (CC) for each comparison are included in the Appendix A and visualized in Figure 4e,f. The results of the GO analysis for the cellular components (CC) in the comparison of significant phosphoproteins between the 2G12 and 353/11 samples indicated significant enrichment in terms related to “nuclear lumen”, “membrane-enclosed lumen”, “organelle lumen”, “intracellular organelle lumen”, “ribonucleoprotein complex, “nucleoplasm”, and “spliceosomal complex”. These findings suggest that the phosphoproteins in the 2G12 sample are involved in processes such as RNA processing and spliceosome assembly. In contrast, TM treatment affected cellular component-related terms, such as “non-membrane-bounded organelle”, “intracellular non-membrane-bounded organelle”, “site of polarized growth”, and “somatodendritic compartment”, often related to compartments where cell growth takes place. In addition, nuclear chromatin, heterochromatin, and perichromatin terms are highlighted after TM treatment, describing the different sites at which tunicamycin acts.

Altogether, GO term analysis identified that the factor difficult to express mAb 2G12 acts predominantly in the transcription machinery of the cellular nucleus, while the factor tunicamycin treatment influences various processes like morphogenesis, growth, the p53 family of proteins, and chromatin structures, either in the cytosol or in the nucleus. All such complex interactions are adequately described by the term “TOR (target of rapamycin) signaling”. TOR and its mammalian ortholog mTOR are serine-threonine kinases that sense growth factors, the cellular nutrient, or oxygen status and promote appropriate changes in cell growth and proliferation, cell survival, and protein synthesis. TOR signaling has key roles in cancer, autophagy, and other cellular pathways.

### 3.3. Dynamics of Phosphorylation in the Two Different CHO Cell Lines That Produce the Antibodies

Network analysis enrichment of terms related to significant (phospho)proteins in the comparison between 2G12 and 353/11 was performed using ShinyGO (version 0.80) (Figure 5). The significant terms are predominantly associated with the nucleus and include a wide range of terms like “basal transcription machinery binding”, “spliceosomal complex”, “ribonucleoprotein complex”, “RNA metabolic process”, “RNA binding”, “nucleic acid metabolic process”, “gene expression”, “nuclear lumen”, “nucleoplasm”, “nucleus”, “intracellular organelle lumen”, “membrane-enclosed lumen”, and “organelle lumen” (Appendix A). A few phosphoproteins are responsible for the significance of these terms and are therefore described in more detail here.

In particular, heterogeneous nuclear ribonucleoprotein K (Hnrnpk) plays a critical role in the regulation of various cellular processes, such as RNA stability, pre-mRNA processing and transport, transcription, translation, and chromatin remodeling [75]. Our data indicate that Hnrnpk is phosphorylated at Ser116 in the 2G12 samples, which has been associated with the activation of downstream signaling pathways that initiate DNA damage repair, leading to cell cycle arrest. It also triggers the export of Hnrnpk protein (ribonucleoprotein complex) from the nucleus to the cytoplasm [76].

The genes responsible for regulating the term “ribonucleoprotein complex” include these proteins: Hnrnpk, Taf2, Nop58, Utp18, Cwc27, Srrm1, Vim, Usp39, and Nolc1. In our study, Nop58, Srrm1, and Usp39 were hypophosphorylated. On the other hand, the phosphorylation of Hnrnpk, Taf2, Utp18, Cwc27, Vim, and Nolc1 was increased.

We conclude that expression of 2G12 regulates nuclear pathways based on observed increases and decreases in phosphorylation abundances. Interestingly, regulation of phosphoprotein sites does not lead to significant differences in the ER stress response. It can be speculated that other regulatory factors, aside from phosphorylation are responsible for the feedback regulation in the nucleus, although it is initiated by the maturing 2G12. Alternatively, some of the identified phosphoproteins, which are also part of the ER, may act as second messengers to modulate the transcriptional machinery.

### 3.4. Function of Ribosomal Proteins, DNA Methylation, and Regulation of Chromatin Affected by Tunicamycin

Tunicamycin-treated samples revealed a more complex picture of phosphorylation-regulated pathways. The key terms describe DNA methylation processes, chromatin dynamics and structure, organized cellular structures with distinct morphology and function, and regulation of autophosphorylations, known as a major factor in the stress response. Notably, the TOR (target of rapamycin) signaling pathway, which plays a critical role in many cytosolic and nuclear regulatory processes (Appendix A) (Appendix A), is also regulated by mild treatment with tunicamycin.

Tunicamycin induces broad phosphoproteomic changes that affect epigenetic regulation, organelle dynamics, and stress response mechanisms. Specifically, phosphoproteins associated with DNA methylation at cytosine residues within CG sequences and chromatin organization suggest that tunicamycin may influence gene expression and epigenetic modulation. Additionally, the enrichment of non-membrane-bound organelles and cytoskeletal components implies a reorganization of cellular structures, likely connected to the activation of the unfolded protein response (UPR) in the context of endoplasmic reticulum (ER) stress.

Following the Gene Ontology (GO) analysis and focusing exclusively on biological processes, the term “cellular component biogenesis” was identified as one of the most significantly enriched terms due to its broad relevance regarding the organization of cellular components (Appendix A). This term encompasses a list of differentially phosphorylated sites from seven proteins: Rps6, Mcm2, Cebpz, Ehmt2, Map1b, Smarca4, and Dbnl. Rps6 (ribosomal protein s6) has been shown to be part of the 40s subunit [77,78] and its phosphorylation is often used as an indicator of mTORC1 activity [79].

Phosphorylation of Rps6 plays a critical role in the regulation of cell size [80], glucose homeostasis [81,82], and the function of the translation machinery in mammalian cells [83,84]. Notably, we identified all five phosphorylated residues documented by [85]: Ser235, Ser236, Ser240, Ser244, and Ser247. Remarkably, the latter three phosphoserines (Ser240, Ser244, and Ser247) are separated by a single non-phosphorylated serine, as reported in the referenced study [85]. The 40S ribosomal subunits have been proposed to play a role in initiating translation of specific mRNAs required for adaptation to stress [86,87], which in our case is tunicamycin-induced. Other ribosomal proteins have also been found to be regulated at the proteomic level after TM treatment [12], including the Fau ribosomal protein S30, a component of the 40S subunit, although its overall abundance exhibited a modest decrease at comparable protein concentrations.

Two other important terms are “DNA methylation” and “DNA alkylation”, which have been identified as regulated key processes. DNA methylation is a critical epigenetic mechanism involved in transcriptional silencing, while DNA alkylation plays a role in genomic regulation. Proteins, such as Ehmt2 and Smarca4, are involved in these processes and are essential for regulating chromatin structure and gene expression through epigenetic modifications [88,89]. Ehmt2, also known as euchromatic histone lysine methyltransferase-2, is a histone methyltransferase that primarily methylates histone H3 at lysine 9 (H3k9me1 and H3k9me2) [90] and is generally associated with transcriptional repression. It also functions as a key regulator of heterochromatin formation and maintenance, contributing to the silencing of specific genes during development [91]. Ehmt2 also interacts with DNA methyltransferases (Dnmts), such as Dnmt3a and Dnmt3b, and facilitates the recruitment of these enzymes to specific genomic loci [92]. Smarca4, a member of the Swi/Snf chromatin remodeling complex, is an ATP-dependent chromatin remodeler that drives changes in nucleosome positioning, thereby regulating DNA accessibility [93]. This remodeling activity is crucial for enabling the recruitment of repair proteins to sites of DNA damage, including those resulting from alkylation, thereby promoting effective DNA repair mechanisms [88].

Whenever the chromatin structure is modified, the protein minichromosome maintenance 2 (Mcm2) is involved, and it was also found to be regulated upon tunicamycin treatment. Enrichment analysis revealed that Mcm2 is associated with enriched processes such as “chromatin” and “chromosome” (Appendix A) (Appendix A). Mcm2 is part of the Mcm2-7 complex, a group of six related proteins that are essential for the initiation and elongation of DNA replication in eukaryotic cells [94]. Furthermore, phosphorylation of Mcm proteins has been shown to be crucial for the initiation of DNA replication in eukaryotic cells [95].

Collectively, these results underscore the multiple effects of tunicamycin on cellular processes and emphasize its role in modulating phosphoproteomic profiles related to epigenetics, organelle function, and stress adaptation mechanisms in a more complex manner than expression of the difficult-to-produce mAb 2G12. Appendix A also highlights the multifaceted effect of tunicamycin, indicating a broad imbalance of cells.

### 3.5. Identification of Phosphorylated Key Proteins Under TM as an Extrinsic Stress Factor

To identify regulated phosphorylation on proteins, corresponding to TM-induced stress, we selected relevant terms from Appendix A, specifically “response to stress”, “cellular response to stress”, and “response to osmotic stress” (highlighted in light blue). This analysis revealed seven phosphoproteins associated with these stress responses, namely: Eef2k, Ehmt2, Hsp90ab1, Map1b, Larp1, Tnks1bp1, and Nolc1, as listed in Table 1.

Eukaryotic elongation factor 2 kinase (**Eef2k**) is a calcium/calmodulin-dependent enzyme that serves as a critical regulator of protein synthesis [96]. Inactivation of Eef2k (often through phosphorylation) promotes protein translation by preventing the phosphorylation of eukaryotic elongation factor 2 (eEF2), thereby allowing eEF2 to remain active and facilitate protein synthesis [97,98]. Eukaryotic elongation factor 2 (eEF2), a monomer, catalyzes the translocation of elongating ribosomes [99] relative to mRNA [100]. Eef2k activity is regulated by multiple signaling pathways, with the mammalian target of rapamycin complex 1 (mTORC1) playing a key role [101]. The mTORC1 promotes the phosphorylation of Eef2k at inhibitory sites, thereby blocking Eef2k activity and leading to no phosphorylation of eEF2 and activation of translational elongation [101,102,103]. One of the commonly used indicators of mTORC1 activity [79], already discussed in this manuscript, is ribosomal protein S6 (Rps6).

In TM-treated samples, we observe that Ser73 on Eef2k is phosphorylated. Under prolonged tunicamycin treatment (minimum 12 h), Eef2k is typically activated [104], which likely serves as a mechanism to inhibit protein synthesis under stressful conditions [97], thereby helping the cell to manage the accumulation of misfolded proteins. We also observed low levels of phosphorylation of Eef2k and speculate that this is part of a more general TM response. This continuation of protein synthesis prepares the cell to subsequently increase the expression of molecular chaperones that facilitate proper protein folding and help alleviate ER stress [105]. This suggests that phosphorylation of eEF2 may interfere with or prevent eEF2 binding to the ribosome, reducing the affinity of eEF2 for ribosome complex formation and ultimately rendering it inactive in the elongation phase of translation [106,107].

Other highly enriched biological processes (BP) terms associated with the increase in phosphorylation for Eef2k include “positive regulation of cell morphogenesis involved in differentiation” and “regulation of protein autophosphorylation”. Additionally, the significantly enriched molecular function (MF) terms include “nucleic acid binding” and “translation regulator activity”. The kinase Eef2k was not detected in our recent proteome paper [12].

Euchromatic histone lysine methyltransferase-2 (**Ehmt2**) has been shown in another paper to modulate the expression of genes related to stress responses, such as those involved in apoptosis or endoplasmic reticulum (ER) stress [108]. The protein was not detected in the list of proteins from our recent LFQ proteomics study [12]. In our data, it is found to be phosphorylated at Ser229; however, no data have been published on this specific phosphorylation site. Conversely, another study [109] has shown that phosphorylation at Ser211 is associated with mediating DNA damage repair.

The decrease in phosphorylation abundance of **Hsp90ab1** (heat shock protein HSP 90-beta) at Ser255 can be seen as strongly associated with the Akt signaling pathway and has been identified as a target of the kinase Tssk4 at this position [110]. This position (Ser255) has also been shown to be crucial in the MAPK/ERK signaling pathway, having an essential role in cell growth and cell viability [111]. Phosphorylation at Ser255 may inhibit Hsp90ab1 ATPase activity, potentially restricting the Pi3k/Akt pathway and leading to decreased survival functions, ultimately resulting in apoptosis of the target cells [110,112]. Akt, also known as protein kinase B (Pkb), has an essential role in preventing apoptosis and promoting cell survival [113,114]. The proteomic data from our recent study [12] detected Hsp90ab1, but it was not classified as a significant protein. Our analysis revealed that among the most significantly enriched BP terms associated with Hsp90ab1 were “response to organonitrogen compound”, “cell development”, and “regulation of cell size”. Furthermore, the significant enriched MF terms include “regulation of cell size”, “DNA polymerase binding”, “protein domain specific binding”, and “UTP binding”.

The observed decrease in phosphorylation of **Map1b** at position Ser1255 might play a role for Map1b in stress response or aging [115]. The proteomic data from our previous study [12] detected the protein, although it was not classified as a significant hit. Changes in microtubules are thought to influence cellular responses to environmental stress, with microtubule-associated proteins playing a role in these stress mechanisms [116]. Additionally, in another study, treatment with a different stressor (rapamycin) resulted in decreased phosphorylation of Map1b at Serine 1265 [117]. The proposed role of phosphorylated Map1b in maintaining microtubule integrity suggests its contribution to keeping microtubules in a dynamically unstable state [118]. We identified that the most significantly enriched BP terms associated with the Map1b include “response to organonitrogen compound” and “regulation of cell size”. In terms of MF, the significantly enriched terms included “protein-containing complex binding”, “cytoskeletal regulatory protein binding”, and “phospholipid binding”. For cellular components (CC) terms, the top hits were “non-membrane-bounded organelle”, “intracellular non-membrane-bounded organelle”, and “site of polarized growth”.

**Larp1** shows an increase in phosphorylation at sites Ser735 and Ser743. Larp1 is a direct substrate of mTORC1, and its phosphorylation by mTORC1 leads to its dissociation from the 5′ untranslated region (UTR) of mRNA, thereby relieving its inhibitory activity on translation [119]. We identified that the most significantly enriched BP terms associated with Larp1 include “response to organonitrogen compound” and “TOR signaling”. Additionally, the significantly enriched MF terms include “RNA binding” and “protein-containing complex binding”. For CC terms, among the top hits were “non-membrane-bounded organelle” and “intracellular non-membrane-bounded organelle”. In our proteome study [12], Larp1 was not detected in the list of proteins.

**Tnks1bp1** (182 kDa tankyrase-1-binding protein), was found to be phosphorylated at Ser1630. Tnks1bp1 is essential for the efficient repair of DNA double-strand breaks and is localized in both the nucleus and the cytoplasm [120]. Furthermore, Tnks1bp1 has been shown to co-immunoprecipitate with Tankyrase 1 (Tnks1) [121], a protein associated with the Wnt/β-catenin signaling pathway [122]. Tnks1bp1 was detected in the proteome study [12]; however, it was not classified as a significant hit. Among the significantly enriched BP terms for Tnks1bp1 were “regulation of protein autophosphorylation” and “cellular nitrogen compound metabolic process”. Enriched MF terms include “protein domain specific binding” and “protein-containing complex binding”. For CC terms, “non-membrane-bounded organelle” and “intracellular non-membrane-bounded organelle” were the most prominent.

**Nolc1** (nucleolar and coiled-body phosphoprotein 1) is hypophosphorylated at position Ser542 compared to non-TM-treated samples. Nolc1 is predominantly found in the nucleus and is thought to play an important role in transcription and translation processes [123]. In our previous proteomics study [12], we observed that tunicamycin (TM) did not influence the Nolc1 protein level. Consequently, we may conclude that the observed effects on Nolc1 are attributable solely to changes in phosphorylation levels. Nolc1 was detected in the proteomic analysis [12], but it was not in the list of significant proteins. The significant enriched BP terms associated with Nolc1 are “cell development” and “cellular nitrogen compound metabolic process”, while the enriched MF terms were “RNA binding”, “protein domain specific binding”, and “nucleic acid binding”. For CC terms, the top hits were “non-membrane-bounded organelle” and “intracellular non-membrane-bounded organelle”.

## 4. Conclusions

The entire proteome encompasses not only proteins but also diverse proteoforms, including those modified by post-translational modifications (such as phosphorylation or glycosylation) and other protein variants within a cell or tissue. This makes the proteome inherently more complex and dynamic than the genome, as each gene has the potential to produce multiple protein forms.

In our approach, we used label-free quantification (LFQ), using quantitative LC-MS/MS to generate consistent data at both proteomics and phosphoproteomics levels, to minimize variability in sample handling. One limitation of this approach lies in the inherent assumption that the identification of one or two closely spaced peptides is indicative of the presence of intact proteins. This reliance may inadvertently overlook the potential for incomplete protein representation or the existence of alternative splicing variants, both of which could significantly impact the observed phosphoproteomic profile. This oversight may lead to an incomplete understanding of the phosphorylation landscape, as noncanonical phosphorylations, such as those on histidine, lysine, aspartate, and glutamate, can play crucial roles in protein function and signaling pathways. Additionally, the study primarily focused on the most abundant phosphorylations, as the quantification method used was based on the total number of MS/MS spectra corresponding to each phosphopeptide. Thus, this approach may have limited the detection of less abundant but potentially significant phosphorylations.

Label-free quantification (LFQ) LC-MS/MS phosphoproteomics is a widely used technique for studying protein phosphorylation dynamics; however, it comes with several limitations. One major disadvantage is the lower reproducibility compared to label-based methods, as signal intensity variability between runs can affect quantification accuracy. This method often suffers from missing values across samples due to stochastic sampling during data-dependent acquisition (DDA), leading to incomplete datasets. LFQ is also highly sensitive to instrument fluctuations and sample preparation inconsistencies, which can compromise data quality. The dynamic range of LC-MS/MS systems poses a challenge when analyzing complex phosphoproteomes, as low-abundance phosphopeptides may be missed, resulting in incomplete coverage. Phosphorylation sites are often substoichiometric, which exacerbates detection issues in label-free approaches. The lack of multiplexing capabilities further limits LFQ’s efficiency in large-scale experiments compared to isotope-labeled methods. Additionally, LFQ often requires extensive data processing, normalization, and computational tools to mitigate technical variability. The need for high sample amounts for reliable quantification can also be a drawback when working with limited biological material. Finally, the time-consuming nature of LFQ workflows, including LC-MS/MS acquisition and subsequent analysis, makes it less suitable for high-throughput studies. In summary, the key disadvantages are missing values, limited sensitivity, variability, and the complexity of data analysis, all of which can affect the accuracy and reproducibility of LFQ LC-MS/MS results.

Our study focused on conducting a phosphoproteomic analysis of CHO-K1 cells to enhance the understanding of complex proteome-based cellular processes affected by the production of the challenging recombinant monoclonal antibody (mAb) 2G12 as an intrinsic cellular factor.

Additionally, the effects of tunicamycin (TM) treatment—as an extrinsic stressor—were examined in the CHO-K1 cells that produce the more easily manufactured mAb 353/11. The analysis identified 2109 proteins and quantified 4059 phosphosites linked to 2504 distinct phosphorylated peptides. The data revealed that serine is the most frequently phosphorylated residue, followed by threonine and tyrosine, which is consistent with previous studies. These findings suggest that phosphorylation is predominantly regulated by serine/threonine kinases, with significant implications for various signaling pathways and biological processes. Although only 26 phosphorylation sites were regulated in TM-treated cells, and 53 in the case of the different antibodies, the number of regulated pathways in TM-treated cells is higher and more complex in terms of the compartment in which the changes take place and the biological processes affected.

The study significantly expands our understanding of the complex CHO-K1 cell phosphoproteome by comparing the production of the hard-to-produce mAb 2G12 with the easy-to-produce mAb 353/11. In particular, proteins associated with the nucleus showed differential phosphorylation between the two samples, especially those involved in RNA processing and gene expression. Heterogeneous nuclear ribonucleoprotein K (Hnrnpk) and other proteins were highlighted for their roles in signaling pathways and stress response. In response to tunicamycin-induced stress, key proteins and phosphorylation sites involved in critical cellular processes, such as mTORC1 signaling, protein synthesis, DNA replication, and stress response, were identified. These findings provide valuable insights into the relatively unexplored field of CHO cell phosphoproteomics, with potential implications for improving cell line engineering and bioprocess optimization. The observed phosphorylation changes suggest a complex regulatory network that enables cells to adapt to external and internal stressors, underscoring potential targets for further research into cellular stress mechanisms and antibody manufacturing.

The findings presented in this study offer critical insights into the phosphoproteomic landscape of CHO-K1 cells during the production of monoclonal antibodies (mAbs) and under tunicamycin-induced stress. However, there remain key opportunities to build upon this work and address its inherent limitations to further advance the understanding of complex cellular processes and improve bioproduction. The gained knowledge will contribute to balance bioprocesses by defining beneficial trace molecules in the medium and provide additional information on feeding strategies and other process conditions resulting in a homeostatic cell population to delay apoptosis.

While our current approach focused on proteomics and phosphoproteomics using LFQ LC-MS/MS, future studies should aim to comprehensively explore diverse proteoforms, including those arising from alternative splicing variants and other post-translational modifications (PTMs), such as glycosylation, acetylation, and ubiquitination. Integrative multi-omic approaches that combine phosphoproteomics with additional layers of protein modification data could provide a holistic understanding of protein function and regulation. Specifically, noncanonical phosphorylations (e.g., on histidine, aspartate, and lysine residues) remain underexplored and may reveal new regulatory mechanisms in stress responses and antibody production.

Future studies could investigate the outcome of phosphorylation events in response to external stressors like tunicamycin. Time-resolved phosphoproteomic analyses would help delineate the sequence of phosphorylation changes and their role in stress adaptation mechanisms. Similarly, subcellular fractionation combined with LC-MS/MS could improve the spatial resolution of phosphoproteomic data, enabling a more precise understanding of compartment-specific regulatory networks, especially in the nucleus and endoplasmic reticulum. The identification of proteins involved in stress response and gene regulation highlights their potential significance in CHO cell biology. Moving forward, functional validation using techniques like CRISPR/Cas9 gene editing, RNA interference, or site-directed mutagenesis of specific phosphorylation sites will be critical for understanding their role in cellular stress adaptation and antibody production. In summary, future efforts should prioritize improving detection sensitivity, expanding the scope of proteoform analysis, and functionally characterizing key proteins and phosphosites. These advancements will not only deepen our understanding of the phosphoproteome but also contribute to the development of more efficient CHO cell lines for biopharmaceutical production. In addition, the gained knowledge will contribute to balance bioprocesses by defining beneficial trace molecules in the medium and provide additional information on feeding strategies resulting in a homeostatic cell population to delay apoptosis.

## Figures and Tables

**Figure 1 proteomes-13-00009-f001:**
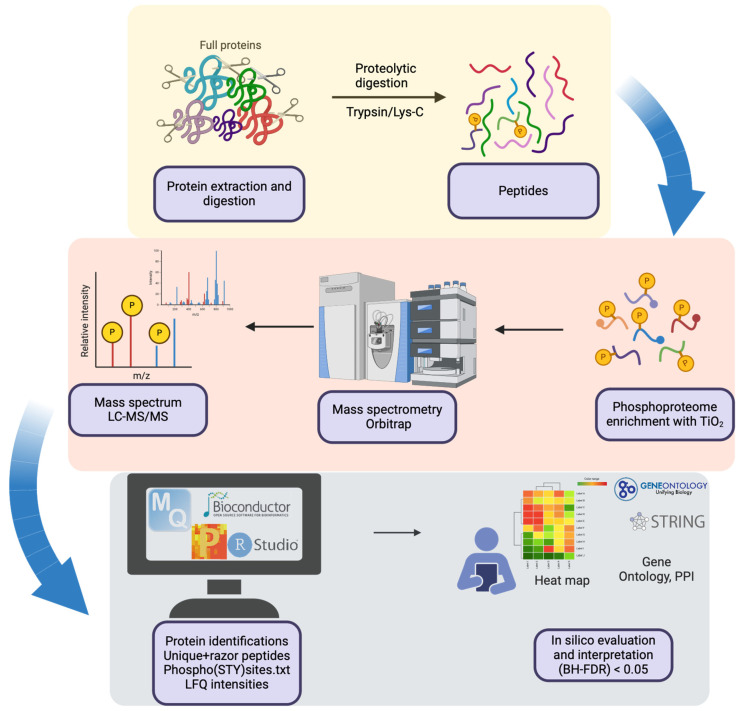
Schematic workflow of label-free quantitative (LFQ) phosphoproteomics coupled with liquid chromatography-tandem mass spectrometry (LC-MS/MS) to analyze phosphorylation levels in proteins derived from Chinese hamster ovary (CHO) cells. The proteins were initially digested with trypsin to generate peptides for further analysis at the proteomics and phosphoproteomics level. Subsequently, phosphopeptides were selectively enriched using a Titanium Dioxide (TiO_2_) Phosphopeptide Enrichment Kit. The phosphorylation levels of these phosphopeptides were then analyzed and compared using a label-free liquid chromatography-tandem mass spectrometry (LC–MS/MS) technique. Created with BioRender.com.

**Figure 2 proteomes-13-00009-f002:**
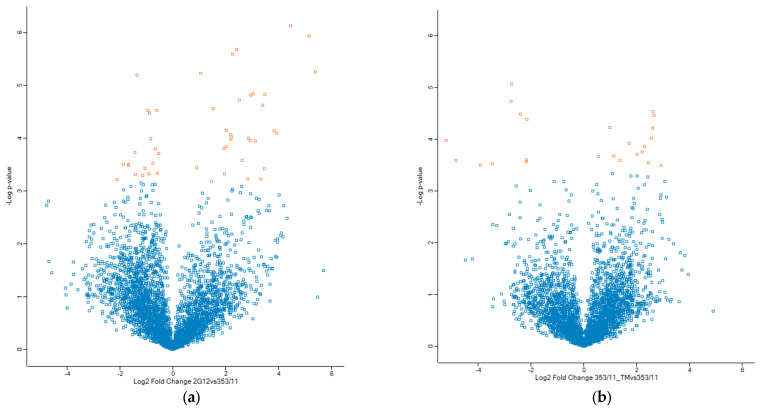
Volcano plots depicting fold changes of proteins adjusted for multiple testing using a permutation-based FDR, with *q* < 0.05. On the X-axis, the Log2 fold change is shown between the groups (**a**) 2G12 vs. 353/11 and (**b**) 353/11_TM vs. 353/11, respectively. The Y-axis portrays −Log10 *p*-values, adhering to the corresponding *q*-value after correcting for multiple hypotheses. Proteins colored in orange exhibit *q* < 0.05. The analysis was conducted using Perseus software.

**Figure 3 proteomes-13-00009-f003:**
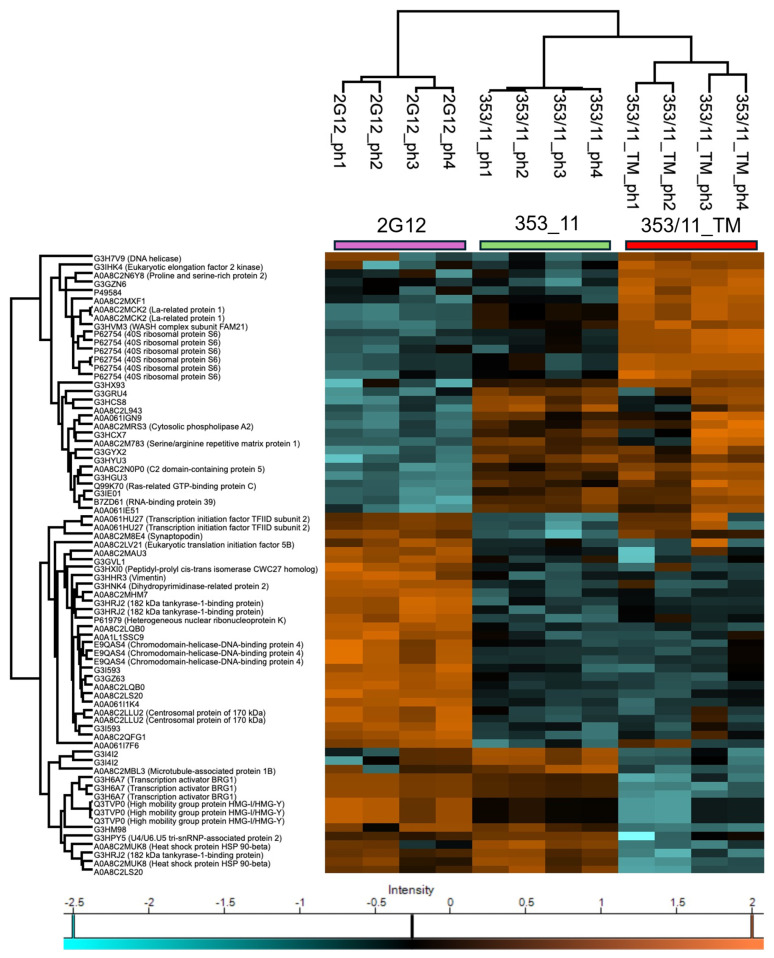
A heatmap, generated using Perseus software, illustrating the expression overview of all significantly different phosphosites (rows) across all samples (columns) between the 2G12 vs. 353/11 and 353/11_TM vs. 353/11 comparisons (BH–FDR, *q*-value < 0.05) (Appendix A). Phosphosites with higher intensities are highlighted in orange, while those with lower intensities are indicated in light blue.

**Figure 4 proteomes-13-00009-f004:**
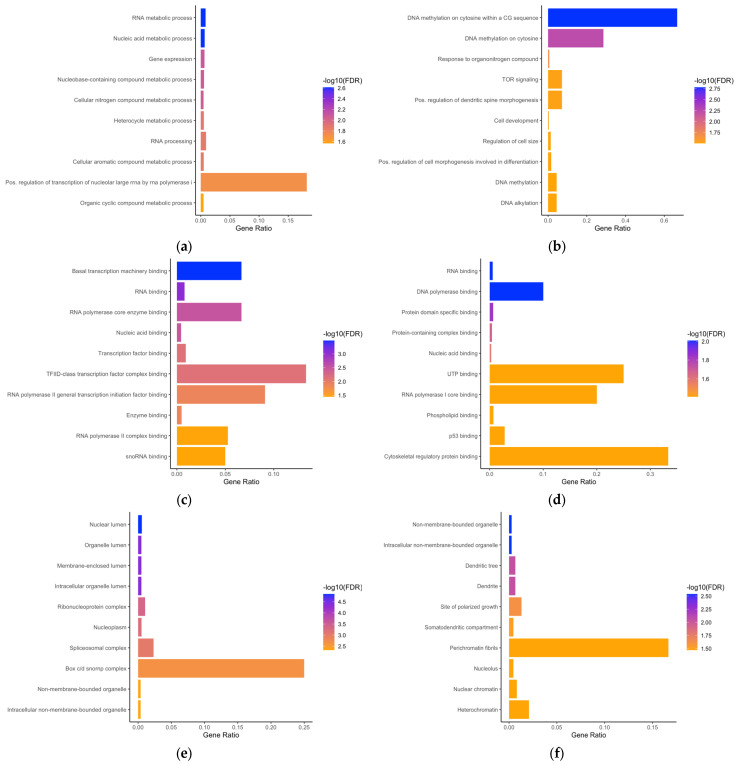
Top Gene Ontology (GO) terms for biological process (BP), molecular function (MF), and cellular compartment (CC) enrichment analysis based on the ShinyGO tool. The length of each bar indicates the gene ratio (nGenes/PathwayGenes) linked to the corresponding enriched term, while the color denotes the statistical significance of the enrichment, ranging from high significance (blue) to low significance (orange): (**a**) the bar chart displays the Gene Ontology (GO) enrichment analysis for biological processes (BP) of differentially expressed phosphoproteins between the 2G12 and 353/11 comparison; (**b**) GO enrichment for biological processes (BP) between 353/11_TM and 353/11; (**c**) GO enrichment for molecular functions (MF) of differentially expressed phosphoproteins between 2G12 and 353/11; (**d**) GO enrichment for molecular functions (MF) between 353/11_TM and 353/11; (**e**) GO enrichment for cellular compartments (CC) of differentially expressed phosphoproteins between 2G12 and 353/11; and (**f**) GO enrichment for cellular compartments (CC) between 353/11_TM and 353/11.

**Figure 5 proteomes-13-00009-f005:**
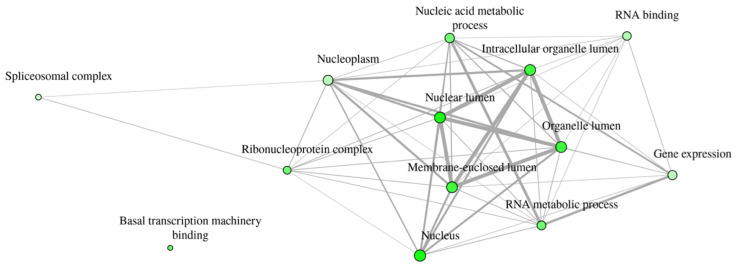
The enrichment plot network analysis comparing 2G12 and 353/11 illustrates the top 13 identified terms across different functional categories, using an edge cut-off of 0.3. Nodes correspond to the enriched terms, with edges representing their interactions. The thickness of the edges reflects the degree of gene overlap, while node size indicates the number of genes associated with each term (Appendix A).

**Table 1 proteomes-13-00009-t001:** The seven proteins comprising the term “response to stress”.

Protein	Protein Names	Gene Names	Sequence Window	Residue	Log2 Change 353/11_TM vs. 353/11
G3IHK4	Eukaryotic elongation factor 2 kinase	Eef2k	NYYSNLMKTECGSTGSPASSFHFKEAWKHAI	S73	2.31
A0A8C2MXF1	euchromatic histone lysine methyltransferase 2 (Ehmt2)	Ehmt2	LGKVTSDAAKRRKLNSGSLSEDFGSARGSGD	S229	1.72
A0A8C2MUK8	Heat shock protein HSP 90-BETA	Hsp90ab1	EDKDDEEKPKIEDVGSDEEDDSGKDKKKKTK	S255	−3.46
A0A8C2MBL3	Microtubule-associated protein 1B; Map1b heavy chain; Map1 light chain Lc1	Map1b	IKDVSDERLSPTKSPSLSPSPPSPIEKTPLG	S1255	−5.19
A0A8C2MCK2	La-related protein 1	Larp1	ANKLFGAPEPSTIARSLPTTVPESPNYRNAR	S735	1.00
			EPSTIARSLPTTVPESPNYRNARTPRTPRTP	S743	1.00
G3HRJ2	182 kDa tankyrase-1-binding protein	Tnks1bp1	LSPSALKAKLRSRNRSAEEGEVTESKSSQKE	S1630	−3.92
A0A8C2LS20	Nucleolar and coiled-body phosphoprotein 1	Nolc1	ANGTPASQNGKAGKESEEDEEEEETKMAVSK	S542	−4.83

## Data Availability

Phosphoroteomics data generated in this study have been deposited via ProteomeXchange, with identifier PXD055200.

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
