# Peer review of "Systems Biology of Recombinant 2G12 and 353/11 mAb Production in CHO-K1 Cell Lines at Phosphoproteome Level"

_proteomes, 2025, doi:10.3390/proteomes13010009_

Round 1
Reviewer 1 Report
Comments and Suggestions for Authors
A phosphoproteomic LFQ approach was applied to three CHO cell lines. 4059 phosphosites were confidently identified from 2109 CHO proteins in total from the 3 cell lines.
How many phosphopeptides were confidently identified in this experiment?
The 2G12 vs 353/11 comparison yielded 53 phosphosites that were regulated? How many phosphopeptides were regulated in this comparison?
The same question for 353/11 vs treated with Tunicamycin and the 26 phosphosites that were regulated.
The total number of regulated phosphosites is very low in my opinion. I am not sure if this indicates a problem with the study as you might expect more phosphopeptides to change in such a study?
The authors describe a prior extensive study of the high producers vs the low producers on the cellular proteome. Overlapping this data could look at the protein ratio, phosphopeptide ratio, and the unmodified peptide counterpart ratio between the samples if the proteins were confidently identified in both studies.
Was the Gene ontology enrichment was performed on the 74 sites and their associated proteins or was it performed on the total of 4059 phosphosites from 2109 proteins? This isn't clear.
Post-translational should be hyphenated in line 51.
The addition of 25% TFA was added to stop the enzymatic reaction (line164) – what volume or what was the final concentration of TFA?
Overall, I am not sure that this study as described by the author was a comprehensive phosphoproteomic analysis (as described in line 693) or that it significantly expands our understanding of complexity of CHO-K1 (described in line 707).
Author Response
Reviewer 1
-A phosphoproteomic LFQ approach was applied to three CHO cell lines. 4059 phosphosites were confidently identified from 2109 CHO proteins in total from the 3 cell lines. How many phosphopeptides were confidently identified in this experiment?
In agreement with your comment, the analysis of the data, after filtration, determined that 2504 phosphopeptides were confidently identified in this experiment by looking at Peptide IDs column in File S1.
We improved the text by addition of the following information (see row 731-734):
“The analysis identified 2109 proteins and quantified 4059 phosphosites linked to 2504 distinct phosphorylated peptides. The data revealed that serine is the most frequently phosphorylated residue, followed by threonine and tyrosine, consistent with previous studies.
-The 2G12 vs 353/11 comparison yielded 53 phosphosites that were regulated? How many phosphopeptides were regulated in this comparison?
By examining the Peptide ID column, it was determined that 41 regulated phosphopeptides were associated with these phosphosites in this comparison.
For better understanding we added this information in the following sentence to the Result section:
(see row 350-354)
“Specifically, 53 significant phosphosites (associated with 41 phosphopeptides) were identified between the 2G12 (low producer) and 353/11 (high producer) groups, while 26 significant phosphosites (associated with 19 phosphopeptides) were found between the 353/11_TM (high producer treated with TM) and 353/11 (high producer) groups (Figure 2) (Table S2).”
-The same question for 353/11 vs treated with Tunicamycin and the 26 phosphosites that were regulated.
Upon further analysis of the "Peptide ID" column, 19 regulated phosphopeptides were associated with these phosphosites.
We added this information in the following sentence to the results section:
(see row 359-363)
“Specifically, 53 significant phosphosites (associated with 41 phosphopeptides) were identified between the 2G12 (low producer) and 353/11 (high producer) groups, while 26 significant phosphosites (associated with 19 phosphopeptides) were found between the 353/11_TM (high producer treated with TM) and 353/11 (high producer) groups (Figure 2) (Table S2).”
-The total number of regulated phosphosites is very low in my opinion. I am not sure if this indicates a problem with the study as you might expect more phosphopeptides to change in such a study?
Thank you for your comment. We know that the low number of regulated phosphosites most probably resulted from the specific experimental conditions like the early 4-hour sampling time and the moderate TM concentrations. Previous studies employing prolonged TM treatment with using higher concentrations have demonstrated that cellular responses, such as apoptosis-related factors, become more pronounced under such conditions [1,2]. However, our objective was to focus on identification of early indicators or precursors of cellular stress.
-The authors describe a prior extensive study of the high producer cells vs the low producer cells on the cellular proteome. Overlapping this data could look at the protein ratio, phosphopeptide ratio, and the unmodified peptide counterpart ratio between the samples if the proteins were confidently identified in both studies.
We agree that comparing the protein, phosphopeptide, and unmodified peptide ratios would be valuable. We performed a proteomics analysis with the same samples prior to the phosphoproteomics analysis. However, not all proteins identified with altered expression in the whole proteome were detected in the phosphoproteome, which limited our ability to make direct comparisons. Additionally, other post-translational modifications can mask phosphorylation sites. These factors restricted our ability to fully overlap the data, but we plan to address these challenges in future studies for more robust comparisons.
-Was the Gene ontology enrichment was performed on the 74 sites and their associated proteins or was it performed on the total of 4059 phosphosites from 2109 proteins? This isn't clear.
The GO enrichment analysis was performed on the 74 regulated phosphosites and their associated proteins.
We added this information in the following sentence in the text:
(see row 284-285)
“We analyzed the gene ontologies of the 74 differentially expressed phosphosites using ShinyGO 0.80.”
-Post-translational should be hyphenated in line 51.
As recommended, the following line has been improved. It now reads as follows: post-translational modification (PTM)
(see row 51)
-The addition of 25% TFA was added to stop the enzymatic reaction (line164) – what volume or what was the final concentration of TFA?
The final concentration of TFA in the solution was adjusted to 0.2% by adding 6.7 µL of 25% TFA to the reaction mixture (total volume of 850 µL). This specific volume as recommended in the manufacturer's protocol was sufficient to effectively quench protease activity, thereby halting the enzymatic reaction.
As recommended, the following sentence has been improved. It now reads as follows:
(see row 168-171)
“Finally, the digestion process was terminated with the addition of 25% trifluoroacetic acid (TFA) (Thermo Fisher Scientific, no. 28904, Rockford, IL, USA) to achieve a final concentration of 0.2% TFA in the mixture.”
-Overall, I am not sure that this study as described by the author was a comprehensive phosphoproteomic analysis (as described in line 693) or that it significantly expands our understanding of complexity of CHO-K1 (described in line 707).
We agree with you in the frame of comprehensiveness. However, the current study is one of the first studies using phosphorylation of the CHO cell proteins as an indicator for metabolic processes during the formation of recombinant antibodies and stress induction. We also have revised the text as recommended and removed the word comprehensive from the sentence:
(See row 725)
(See row 755-762)
References:
- Chandrawanshi, V.; Kulkarni, R.; Prabhu, A.; Mehra, S. Enhancing titers and productivity of rCHO clones with a combination of an optimized fed-batch process and ER-stress adaptation. J Biotechnol 2020, 311, 49-58, doi:10.1016/j.jbiotec.2020.02.008.
- Zhang, C.; Fu, Y.; Zheng, W.; Chang, F.; Shen, Y.; Niu, J.; Wang, Y.; Ma, X. Enhancing the Antibody Production Efficiency of Chinese Hamster Ovary Cells through Improvement of Disulfide Bond Folding Ability and Apoptosis Resistance. Cells 2024, 13, doi:10.3390/cells13171481.
Reviewer 2 Report
Comments and Suggestions for Authors
This study investigated the phosphoproteomics of recombinant 2G12 and 353/11 mAb production in 2 CHO-K1 cell lines. The results significantly expands our understanding of the complex CHO-K1 cell phosphoproteome by comparing the production of the hard-to-produce mAb 2G12 with the easy-to-produce mAb 353/11. I have a few comments that I hope the authors will consider.
1. The manuscript lacks a clear statement of the primary objective of the study. A more explicit aim should be articulated in the introduction.
2. The literature review is insufficient. More recent studies should be cited to provide context and support for the research question.
3. The methods section is vague in certain areas, particularly regarding LC-MS/MS analysis. More detailed descriptions of experimental procedures are necessary for reproducibility.
4. The discussion section is superficial. Expand on how findings relate to existing literature and their implications for future research.
5. The manuscript contains numerous grammatical errors and awkward phrasing that hinder clarity. A thorough proofreading is required.
6. Limitations of the study are not discussed adequately; acknowledging potential weaknesses strengthens scientific rigor.
7. Suggestions for future research are lacking; propose specific avenues for further investigation based on your findings.
The manuscript contains numerous grammatical errors and awkward phrasing that hinder clarity. A thorough proofreading is required.
Author Response
Reviewer 2
This study investigated the phosphoproteomics of recombinant 2G12 and 353/11 mAb production in 2 CHO-K1 cell lines. The results significantly expands our understanding of the complex CHO-K1 cell phosphoproteome by comparing the production of the hard-to-produce mAb 2G12 with the easy-to-produce mAb 353/11. I have a few comments that I hope the authors will consider.
The manuscript lacks a clear statement of the primary objective of the study. A more explicit aim should be articulated in the introduction.
As recommended, we have added the requested information and improved the following text:
(see row 72-75)
“The main aim of our study was to explore the phosphorylation status among CHO cells producing recombinant difficult-to-produce 2G12 vs. easy-to-produce 353/11, as well as examining the influence of the extrinsic stressor tunicamycin (TM) on the protein expression.”
The literature review is insufficient. More recent studies should be cited to provide context and support for the research question.
We have made every effort to ensure that the literature review is extensive and thorough.
(see row 48-49)
“To our knowledge, there are a limited number of published studies on phosphoproteomics approaches in CHO cells [10,13,19-22]”
The methods section is vague in certain areas, particularly regarding LC-MS/MS analysis. More detailed descriptions of experimental procedures are necessary for reproducibility.
Thank you for your comment. We have added information, and the updated section now reads as follows:
(see row 205-206)
“The dried phosphopeptide samples were resuspended in 25 μL of 0.1% formic acid (FA), LC-MS grade (Thermo Fisher Scientific, Cat. No. 85170, Rockford, IL, USA).”
The discussion section is superficial. Expand on how findings relate to existing literature and their implications for future research.
Thank you for your insight. We have expanded on this point with a more detailed explanation. The newly added section in the Discussion now reads:
(see row 755-7771)
“The findings presented in this study offer critical insights into the phosphoproteomic landscape of CHO-K1 cells during the production of monoclonal antibodies (mAbs) and under tunicamycin-induced stress. However, there remain key opportunities to build upon this work and address its inherent limitations to further advance the understanding of complex cellular processes and improve bioproduction. The gained knowledge will contribute to balance bioprocesses by defining beneficial trace molecules in the medium and provide additional information on feeding strategies and other process conditions resulting in a homeostatic cell population to delay apoptosis. While our current approach focused on proteomics and phosphoproteomics using LFQ LC-MS/MS, future studies should aim to comprehensively explore diverse proteoforms, including those arising from alternative splicing variants and other post-translational modifications (PTMs) such as glycosylation, acetylation, and ubiquitination. Integrative multi-omics approaches that combine phosphoproteomics with additional layers of protein modification data could provide a holistic understanding of protein function and regulation. Specifically, noncanonical phosphorylations (e.g., on histidine, aspartate, and lysine residues) remain underexplored and may reveal new regulatory mechanisms in stress responses and antibody production.”
The manuscript contains numerous grammatical errors and awkward phrasing that hinder clarity. A thorough proofreading is required.
Accomplished as recommended.
Limitations of the study are not discussed adequately; acknowledging potential weaknesses strengthens scientific rigor.
We have addressed the reviewer’s comment, and it now reads as follows”
(see row 705-724)
“Label-Free Quantification (LFQ) LC-MS/MS phosphoproteomics is a widely used technique for studying protein phosphorylation dynamics; however, it comes with several limitations. One major disadvantage is the lower reproducibility compared to label-based methods, as signal intensity variability between runs can affect quantification accuracy. This method often suffers from missing values across samples due to stochastic sampling during data-dependent acquisition (DDA), leading to incomplete datasets. LFQ is also highly sensitive to instrument fluctuations and sample preparation inconsistencies, which can compromise data quality. The dynamic range of LC-MS/MS systems poses a challenge when analyzing complex phosphoproteomes, as low-abundance phosphopeptides may be missed, resulting in incomplete coverage. Phosphorylation sites are often substoichiometric, which exacerbates detection issues in label-free approaches. The lack of multiplexing capabilities further limits LFQ's efficiency in large-scale experiments compared to isotope-labeled methods. Additionally, LFQ often requires extensive data processing, normalization, and computational tools to mitigate technical variability. The need for high sample amounts for reliable quantification can also be a drawback when working with limited biological material. Finally, the time-consuming nature of LFQ workflows, including LC-MS/MS acquisition and subsequent analysis, makes it less suitable for high-throughput studies. In summary, the key disadvantages are missing values, limited sensitivity, variability, and the complexity of data analysis, all of which can affect the accuracy and reproducibility of LFQ LC-MS/MS results.”
Suggestions for future research are lacking; propose specific avenues for further investigation based on your findings.
Thank you for your comment. We have now explained it in more detail. The new added part reads as follows:
(see row 772-790)
“Future studies could investigate the of phosphorylation events in response to external stressors like tunicamycin. Time-resolved phosphoproteomic analyses would help delineate the sequence of phosphorylation changes and their roles in stress adaptation mechanisms. Similarly, subcellular fractionation combined with LC-MS/MS could improve the spatial resolution of phosphoproteomic data, enabling a more precise understanding of compartment-specific regulatory networks, especially in the nucleus and endoplasmic reticulum. The identification of proteins involved in stress response and gene regulation highlights their potential significance in CHO cell biology. Moving forward, functional validation using techniques like CRISPR/Cas9 gene editing, RNA interference, or site-directed mutagenesis of specific phosphorylation sites will be critical for understanding their roles in cellular stress adaptation and antibody production. In summary, future efforts should prioritize improving detection sensitivity, expanding the scope of proteoform analysis, and functionally characterizing key proteins and phosphosites. These advancements will not only deepen our understanding of the phosphoproteome but also contribute to the development of more efficient CHO cell lines for biopharmaceutical production. Besides, the gained knowledge will contribute to balance bioprocesses by defining beneficial trace molecules in the medium and provide additional information on feeding strategies resulting in a homeostatic cell population to delay apoptosis.”
Reviewer 3 Report
Comments and Suggestions for Authors
A nice study of comparative study on phosphorylation of cell proteome is presented. CHO-K1 cell producing a difficult versus an easy to manufacture mAb were compared, and cells under tunicamycin treatment were compared with cells without treatment. Despite the extensive study on the differential expression of phosphorylated patterns observed it is not clear how much of these effects are due to the variable being study versus other biological differences such as cell growth rate.
Specific comments:
- There is important information on cell growth and when samples were taken missing on the results section. Were the cells under different conditions growing at similar growth rates?, were the cells under similar cell densities? What about viability? Samples were taken under exponential of stationary growth?
- It is also missing the information on cell productivity, how much mAb were the cells producing? How much difference in product titre there is between the difficult to express proteins and the easy to express one?
- How other variables that could affect the differences in phosphorylation were taken into account in this study?
In figure 3 the text on the name of proteins identified with differential phosphorylation expression is impossible to read, as part of some of the names are under the heatmap.
Author Response
Reviewer 3
A nice study of comparative study on phosphorylation of cell proteome is presented. CHO-K1 cell producing a difficult versus an easy to manufacture mAb were compared, and cells under tunicamycin treatment were compared with cells without treatment. Despite the extensive study on the differential expression of phosphorylated patterns observed it is not clear how much of these effects are due to the variable being study versus other biological differences such as cell growth rate.
Specific comments:
- There is important information on cell growth and when samples were taken missing on the results section. Were the cells under different conditions growing at similar growth rates?, were the cells under similar cell densities? What about viability? Samples were taken under exponential of stationary growth?
The cells were cultured by semi-perfusion fermentation with daily medium exchange and monitored for cell counts and viability. We have decided for this method to emphasize growing at similar rates. This is supported by the fact that both cell lines reached maximum density and exhibited sustained viability by day six.
In terms of cell densities, the initial seeding density was 5 x 10^6 cells per mL, and samples were taken after the medium exchange on day six when both cell lines had reached maximum density. Therefore, at the time of sampling, the cells were under similar cell densities.
Viability was closely monitored during the culture process using a Vi-Cell XR Cell Counter, ensuring that both cell lines maintained sustained viability at the sampling point.
As for the growth phase at the time of sampling, the cells were taken four hours after the medium exchange on day six, when both cell lines had reached their maximum density and exhibited sustained viability. Thus, the cells were in the stationary phase when samples were collected for proteomics analysis.
- We noted the that all the extra data illustrating the specific conditions regarding the cells are thoroughly presented in a previously published paper from our group (Schwaigerlehner et al. 2019). The data about viability, cell counts, and time are presented there in detail.
(see rows 123-125)
“As described in [12,51], the cells were grown in suspension culture using a semi-perfusion process [52,53] in 50 mL vent cap spin tubes (Corning, no. 431720, Corning, NY, USA) within an ISF-X shaker…”
- It is also missing the information on cell productivity, how much mAb were the cells producing? How much difference in product titre there is between the difficult to express proteins and the easy to express one?
Regarding cell productivity, the specific monoclonal antibody (mAb) production levels by the cells are not directly stated in the current text. To obtain this information, we added the following information in the text:
(see row 132-133)
“The data highlighting these differences are described in a prior publication from our research group [51].”
- How other variables that could affect the differences in phosphorylation were taken into account in this study?
This was also our main concern, because of numerous intrinsic and extrinsic factors (cell cycle phase, energy status, internal and external signaling stimuli, and mechanical stress induction (shear stress and tension) etc. Therefore, to obtain the reproducibility of the samples, we used a semi-perfusion[1,2] approach, to cultivate the cells exposed to identical conditions, and at similar cell counts and viability (Schwaigerlehner et al. 2019). The same samples are then used for different omics analysis.
-In figure 3 the text on the name of proteins identified with differential phosphorylation expression is impossible to read, as part of some of the names are under the heatmap.
(see Figure 3)
We have improved Figure 3 as reviewer recommended.
References:
- Mayrhofer, P.; Reinhart, D.; Castan, A.; Kunert, R. Rapid development of clone-specific, high-performing perfusion media from established feed supplements. Biotechnol Prog 2020, 36, e2933, doi:10.1002/btpr.2933.
- Mayrhofer, P.; Castan, A.; Kunert, R. Shake tube perfusion cell cultures are suitable tools for the prediction of limiting substrate, CSPR, bleeding strategy, growth and productivity behavior. Journal of Chemical Technology & Biotechnology 2021, 96, 2930-2939, doi:https://doi.org/10.1002/jctb.6848.
Round 2
Reviewer 1 Report
Comments and Suggestions for Authors
The importance of examining the protein expression data in the authors previous study and overlapping with the phosphopeptide measurement in this study is that phosphopeptide measurements are a composite of protein expression and phosphorylation stoichiometry differences (Wu et al 2011).
To rule out the variances from protein abundance differences and to better reflect the impact on phosphorylation status some sort of normalisation of phosphorylation quantification with protein quantification should be looked at by the author in the 7 proteins that they are referring to as changing in phosphorylation abundance i.e. Eukaryotic elongation factor 2 kinase (Eef2k), Euchromatic histone lysine methyltransferase 2 (Ehmt2), Heat shock protein HSP 90-beta (Hsp90ab1), Microtubule-associated protein 1B (Map1b), La ribonucleoprotein 1, translational regulator (Larp1), 182 kDa tankyrase-1-binding protein (Tnks1bp1) and Nucleolar and coiled-body phosphoprotein 1 (Nolc1).
Are these proteins changing in expression in the previous study?
I believe this is crucial for correct interpretation of the phosphorylation quantification data in this paper.
Ref: Wu R, Dephoure N, Haas W, Huttlin EL, Zhai B, Sowa ME, Gygi SP. Correct interpretation of comprehensive phosphorylation dynamics requires normalization by protein expression changes. Mol Cell Proteomics. 2011 Aug;10(8):M111.009654. doi: 10.1074/mcp.M111.009654. Epub 2011 May 7. PMID: 21551504; PMCID: PMC3149096.
Author Response
Reviewer 1
The importance of examining the protein expression data in the authors previous study and overlapping with the phosphopeptide measurement in this study is that phosphopeptide measurements are a composite of protein expression and phosphorylation stoichiometry differences (Wu et al 2011).
To rule out the variances from protein abundance differences and to better reflect the impact on phosphorylation status some sort of normalisation of phosphorylation quantification with protein quantification should be looked at by the author in the 7 proteins that they are referring to as changing in phosphorylation abundance i.e. Eukaryotic elongation factor 2 kinase (Eef2k), Euchromatic histone lysine methyltransferase 2 (Ehmt2), Heat shock protein HSP 90-beta (Hsp90ab1), Microtubule-associated protein 1B (Map1b), La ribonucleoprotein 1, translational regulator (Larp1), 182 kDa tankyrase-1-binding protein (Tnks1bp1) and Nucleolar and coiled-body phosphoprotein 1 (Nolc1).
Are these proteins changing in expression in the previous study?
I believe this is crucial for correct interpretation of the phosphorylation quantification data in this paper.
Thank you for your comment. We have added the following information into the results and discussion part:
(see row 642-643)
In TM-treated samples, we observe that Ser73 on Eef2k is phosphorylated. Under prolonged tunicamycin treatment (minimun 12h), Eef2k is typically hypophosphorylated [102] which likely serves as a mechanism to inhibit protein synthesis under stressful conditions [95], thereby helping the cell to manage the accumulation of misfolded proteins. We also observed low levels of phosphorylation of Eef2k and speculate that this is part of a more general TM response. This continuation of protein synthesis prepares the cell to subsequently increase the expression of molecular chaperones that facilitate proper protein folding and help alleviate ER stress [103]. This suggests that phosphorylation of eEF2 may interfere with or prevent eEF2 binding to the ribosome, reducing the affinity of eEF2 for ribosome complex formation and ultimately rendering it inactive in the elongation phase of translation [104,105].
Other highly enriched biological processes (BP) terms associated with the increase in phosphorylation for Eef2k include “positive regulation of cell morphogenesis involved in differentiation” and “regulation of protein autophosphorylation”. Additionally, the significantly enriched molecular function (MF) terms include “nucleic acid binding” and “translation regulator activity”. The kinase Eef2k was not detected in our recent proteome paper [12].
(see row 646-647)
Euchromatic histone lysine methyltransferase-2 (Ehmt2) in another paper has been shown to modulate the expression of genes related to stress responses, such as those involved in apoptosis or endoplasmic reticulum (ER) stress [106]. The protein was not detected in the list of proteins from our recent LFQ proteomics study [12]. In our data, it is found to be phosphorylated at Ser229; however, no data has been published on this specific phosphorylation site. Conversely, another study [107] has shown that phosphorylation at Ser211 is associated with mediating DNA damage repair.
(see row 659-660)
The decrease in phosphorylation abundance of Hsp90ab1 (Heat shock protein HSP 90-beta) at Ser255 can be seen as strongly associated with the Akt signaling pathway and has been identified as a target of the kinase Tssk4 at this position [108]. This position (Ser255) has also been shown to be crucial in the MAPK/ERK signaling pathway, having an essential role in cell growth and cell viability [109]. Phosphorylation at Ser255 may inhibit Hsp90ab1 ATPase activity, potentially restricting the Pi3k/Akt pathway and leading to decreased survival functions, ultimately resulting in apoptosis of the target cells [108,110]. Akt, also known as protein kinase B (Pkb), has an essential role in preventing apoptosis and promoting cell survival [111,112]. The proteomic data from our recent study [12] detected Hsp90ab1, but it was not classified as a significant protein. Our analysis revealed that among the most significantly enriched BP terms associated with Hsp90ab1 included “response to organonitrogen compound”, “cell development” and “regulation of cell size”. Furthermore, the significant enriched MF terms include “Regulation of cell size”, “DNA polymerase binding”, “Protein domain specific binding” and “UTP binding”.
(see row 668-669)
The observed deacrease in phosphorylation of Map1b at position Ser1255 might play a role for Map1b in stress response or aging [113]. The proteomic data from our previous study [12] detected the protein, although it was not classified as a significant hit. Changes in microtubules are thought to influence cellular responses to environmental stress, with microtubule-associated proteins playing a role in these stress mechanisms [114]. Additionally, in another study, treatment with different stressor (rapamycin) resulted in decreased phosphorylation of Map1b at Serine 1265 [115]. The proposed role of phosphorylated Map1b in maintaining microtubule integrity suggests its contribution to keeping microtubules in a dynamically unstable state [116]. We identified that among the most significantly enriched BP terms associated with the Map1b include “response to organonitrogen compound” and “regulation of cell size”. In terms of MF, the significantly enriched terms included “protein-containing complex binding”, “cytoskeletal regulatory protein binding” and “phospholipid binding”. For cellular components (CC) terms, the top hits were “non-membrane-bounded organelle”, “intracellular non-membrane-bounded organelle” and “site of polarized growth”.
(see row 689-690)
Larp1 shows an increase in phosphorylation at sites Ser735 and Ser743. Larp1 is a direct substrate of mTORC1, and its phosphorylation by mTORC1, leads to its dissociation from the 5' untranslated region (UTR) of mRNA, thereby relieving its inhibitory activity on translation [117]. We identified that the most significantly enriched BP terms associated with Larp1 include “response to organonitrogen compound” and “TOR signaling”. Additionally, the significantly enriched MF terms include “RNA binding” and “protein-containing complex binding”. For CC terms, among the top hits were “non-membrane-bounded organelle” and “intracellular non-membrane-bounded organelle”. In our proteome study [12], Larp1 was not detected in the list of proteins.
(see row 695-696)
Tnks1bp1 (182 kDa tankyrase-1-binding protein), was found to be phosphorylated at Ser1630. Tnks1bp1 is essential for the efficient repair of DNA double-strand breaks and is localized in both the nucleus and the cytoplasm [118]. Furthermore, Tnks1bp1 has been shown to co-immunoprecipitate with Tankyrase 1 (Tnks1) [119] a protein associated with the Wnt/β-catenin signaling pathway [120]. Tnks1bp1 was detected in the proteome study [12], however, it was not classified as a significant hit. Among the significantly enriched BP terms for Tnks1bp1 were “regulation of protein autophosphorylation” and “cellular nitrogen compound metabolic process”. Enriched MF terms include “protein domain specific binding” and “protein-containing complex binding”. For CC terms, “non-membrane-bounded organelle” and “intracellular non-membrane-bounded organelle” were the most prominent.
(see row 708-709)
Nolc1 (Nucleolar and coiled-body phosphoprotein 1) is hypophosphorylated at position Ser542 compared to non-TM treated samples. Nolc1 is predominantly found in the nucleus and is thought to play an important role in transcription and translation processes [121]. In our previous proteomics study [12], we observed that tunicamycin (TM) did not influence the Nolc1 protein level. Consequently, we may conclude that the observed effects on NolcI are attributable solely to changes in phosphorylation levels. Nolc1 was detected in the proteomic analysis [12], but it was not in the list of significant proteins. The significant enriched BP terms associated with Nolc1 are “cell development” and “cellular nitrogen compound metabolic process”, while the enriched MF terms were “RNA binding”, “protein domain specific binding” and “nucleic acid binding”. For CC terms, the top hits were “non-membrane-bounded organelle” and “intracellular non-membrane-bounded organelle”.
Reviewer 2 Report
Comments and Suggestions for Authors
Accept
Author Response
Dear Reviewer 2
Thank you for your considerations and acceptance for our paper.
Best Regards
Reviewer 3 Report
Comments and Suggestions for Authors
The presented concerns were clarified in the authors responses. However, no clear changes were made to the manuscript to clarify to the reader of the manuscript the conditions on which the experiment took place and why semi-perfusion conditions were used.
Even if this analysis is building up in a previous published experiment, it is important to state in this manuscript that at day 6 cells reached maximum cell density and which density and viability the cells reached and that day 6 was the sampling day. (I would include figures on cell growth and viability on supplementary files)
Similarly, the data of productivity should be included in supplementary files.
Regarding other factors that could affect the proteomics data, were all cell growing at the same growth rate? This is also an important factor that could affect the proteome.
Author Response
Reviewer 3
The presented concerns were clarified in the authors responses. However, no clear changes were made to the manuscript to clarify to the reader of the manuscript the conditions on which the experiment took place and why semi-perfusion conditions were used.
Even if this analysis is building up in a previous published experiment, it is important to state in this manuscript that at day 6 cells reached maximum cell density and which density and viability the cells reached and that day 6 was the sampling day. (I would include figures on cell growth and viability on supplementary files)
Similarly, the data of productivity should be included in supplementary files.
We understand your concern regarding the data on cell growth, viability and productivity (Schwaigerlehner et al. 2019). However, due to copyright restrictions, it is not possible to include the figures and data in a supplementary file that is publicly accessible.
Regarding other factors that could affect the proteomics data, were all cell growing at the same growth rate? This is also an important factor that could affect the proteome.
That was also our main concern, therefore we used this type of cultivation method to generate cells at very similar cell growth and viability (overlapping values).
We have expanded on this point as you suggested with a more detailed explanation.
(see row 123-150)
As described in [12,51], cells were seeded at an initial density of 5 × 10⁶ cells/mL and grown in suspension culture using a semi-perfusion method with daily medium re-freshment by pelleting of cells and adding new medium [52,53]. Cells were cultured in 50 mL vent cap spin tubes (Corning, No. 431720, Corning, NY, USA) in an ISF-X shaker (Kühner, Basel, Switzerland) at 37°C, 80% humidity, 5% CO2 and 220 rpm. We used this downscaling procedure to simulate a continuous biological process, the ‘healthiest’ and most natural cultivation conditions for biological systems.
In general, the perfusion bioprocess is a cultivation method that promotes physio-logical conditions close to those of the animal organism by supplying nutrients and removing negative waste products such as lactate and ammonium. Although many studies have attempted to investigate the difference between exponential and steady-state growth phases in batch cultures, the influence of extrinsic factors resulting from the non-physiological conditions still remains evident. We have shown in two papers that extrinsic conditions have a massive effect on phenotype [54] and gene transcription [55]. The fact that animal cells are able to change their transcriptional behavior in a moderate time frame led us to an experimental approach in which cells were harvested four hours after the daily media change to ensure physiological homeostasis and to guarantee the same extrinsic conditions for tunicamycin-treated and untreated cells. The conditions of tunicamycin treatment (only 4 hours at a low concentration) were chosen to identify differences that play a role in the initiation-phase of the stress response while avoiding the induction of a stronger ER stress response which is found to be characterized by the pronounced upregulation of well-known proteins, including Hsp90b1, Pdia4, and Grp78 (Hspa5) [56]. This approach is expected to facilitate the identification of early regulatory mechanisms, in contrast to the strong ER stress responses induced by higher tunicamycin concentrations. Both cell lines achieved maximum cell density and sustained viability by day six. Samples were collected four hours after the medium exchange on that day, when cells had reached maximum density.
Schwaigerlehner L, Mayrhofer P, Diem M, Steinfellner W, Fenech E, Oostenbrink C, Kunert R (2019) Germinality does not necessarily define mAb expression and thermal stability. Appl Microbiol Biotechnol 103 (18):7505-7518. doi:10.1007/s00253-019-09998-3